

# New molecular evidence for surface and sub-surface soil erosion controls on the composition of stream DOM during storm events

Marie Denis[1], Laurent Jeanneau[1], Patrice Petitjean[1], Anaëlle Murzeau[1], Marine Liotaud[1], Louison Yonnet[1], Gérard Gruau[1]

[1]Géosciences Rennes UMR 6118, Université de Rennes 1/CNRS, 35042 Rennes, France

*Correspondence to*: Marie Denis (mariedenis57@hotmail.fr)

**Abstract.** Storm events are responsible for more than 60% of the export of dissolved organic matter (DOM) from headwater catchments due to an increase in both the discharge and concentration. The latter was attributed to changing water pathways inducing the mobilization of DOM from the surface soil horizons. Recent molecular investigations have challenged this view
and hypothesized (i) a contribution of an in-stream partition of organic matter (OM) between eroded particles and the dissolved fraction and (ii) the modification of the composition of soil DOM during storm events. To investigate these assumptions, soil solutions in the macropores, surface runoff and stream outlet were sampled at high frequency during three storm events in the Kervidy-Naizin catchment, part of the French critical zone observatory AgrHys. The molecular composition of the DOM was analysed by thermally assisted hydrolysis and methylation (THM) with tetramethylammonium hydroxide (TMAH) coupled to
a gas chromatograph and a quadrupole mass spectrometer. These analyses highlighted a modification of the DOM composition in soil solution controlled by the water-table dynamic and pre-event hydrological conditions. These findings fits with the mechanism of colloidal and particulate destabilization in the soil macroporosity. The different behaviour observed for lignins, carbohydrates and fatty acids highlights a potential chemical segregation based on their hydrophobicity. The composition of surface runoff DOM is similar to the DOM composition in soil solution and could be generated by the same mechanism. The
DOM composition in both soil solution and surface runoff corresponds to the stream DOM composition observed during storm events. On the basis of these results, modifications of the stream DOM composition during storm events seem to be due to surface and sub-surface soil erosion rather than in-stream production.

## 1 Introduction

The transfer of organic carbon from soils to rivers and finally to oceans represents an important part of the global carbon cycle.
This organic carbon is transferred as particulate organic matter (POM) and dissolved organic matter (DOM) (Ludwig et al., 1996; Schlesinger and Melack, 1981) which is the most active form of soil organic matter (SOM). The DOM dynamic is highly studied by the scientific community; however, uncertainties remain as to the parameters and processes that control their production, interactions and transfer from soils to aquatic systems (McDowell, 2003).



When considering DOM fluxes at the catchment scale, headwater catchments are the major producer of DOM per surface unit (Ågren et al., 2007). Within catchments, wetlands and riparian zones mostly contribute to DOM export due to high soil organic carbon contents in the first horizons, hydrological connections and extended flooded period which allow water to circulate from soils to rivers (Aitkenhead et al., 1999; Dosskey and Bertsch, 1994; Eckhardt and Moore, 1990). The intensity of DOM

export varies over seasons and hydrological conditions depending on the sources and water flow paths. When considering annual river DOM fluxes, more than 50% is exported during storm events (Buffam et al., 2001; Morel et al., 2009; Raymond and Saiers, 2010). This largest export is attributed to the rise of the water table in organic-rich soil horizons which become hydrologically connected to the river (Boyer et al., 1996), and to the fact that river discharge is mostly sustained by soil water fluxes during storm events (Hagedorn et al., 2000; Lambert et al., 2011). Storm events are also responsible for the modification

of the dissolved organic carbon (DOC) concentration and fluxes in soils. A dilution effect was evidenced by Easthouse et al., (1992) during high precipitation events in organic horizons. Low contact time between the water and soil matrix also creates non-equilibrium situations which lead to a decrease in the DOC concentration in soil solutions (McDowell and Wood, 1984; Michalzik and Matzner, 1999). For all of these reasons, storm events represent special events that must be taken into account in order to provide new insights regarding production mechanisms and the transfer of DOM from a terrestrial to an aquatic

environment.

During the base flow period, DOM is assumed to be produced in the soil microporosity and transferred to the soil macroporosity by diffusion processes, and then to the stream by soil water flow (Worrall et al., 2008). With the establishment of storm flow, the transfer of DOM from the soil solution is assumed to be made conservatively via a piston like effect, as used for the end-member mixing analysis and isotopic studies. However recent studies have demonstrated that this assumption about the

conservative transport of DOM could be impacted by production mechanisms activated during storm flow which could possibly induce a modification of the DOM composition (Dalzell et al., 2005; Hernes et al., 2008). Molecular analyses of lignin biomarkers in stream water DOM have shown that the establishment of storm flow was responsible for an increase in the lignin concentrations and changes in the lignin composition where the lignin in streams during storm flows is less biodegraded compared to the base flow conditions. These modifications, correlated with the turbidity, were attributed to the

activation of new production processes via a chemical equilibrium between the water and POM brought to the river by soils and river bank erosions and which is called an "in-stream process". However, high frequency sampling of stream water during storm flow has shown that the modification of the molecular composition of lignin was persistent even after the decrease of the discharge and turbidity (Jeanneau et al., 2015). To explain these results, other production processes have to be taken into account. One hypothesis that has been proposed is a modification of the DOM composition in soil solutions during storm flow

due to sub-surface erosion which corresponds to SOM erosion in soil macropores triggered by the increase in the water flow velocity (Jeanneau et al., 2015).

Within this context of DOM modification during storm flow, DOM characterization is mostly done using spectroscopic techniques which allow a global characterization. The application of thermally assisted hydrolysis and methylation (THM) using tetramethylammonium hydroxide (TMAH) coupled to a gas chromatograph and mass spectrometer (THM-GC-MS)



appears to be a suitable technique to provide a more precise qualitative characterization of the DOM composition variations. This technique has the advantage of being able to simultaneously analyse biomarkers from lignins (LIG), carbohydrates (CAR) and fatty acids (FA) (Grasset et al., 2009). While LIG come from a plant origin only, CAR and FA could be used to differentiate between a plant and microbial origin. Moreover, the investigation into their distribution may provide new evidence of

compositional modification during the establishment of storm flow.

The main questions investigated are (i) is the DOM composition in soil solution modified during storm flow? (ii) If these changes are observed, are they consistent with modifications in the DOM composition in stream water during storm flow? (iii) Which mechanisms could explain these compositional modifications? To this end, soil solutions and stream water were sampled simultaneously at high frequency during the storm flow period and analysed for their molecular composition. This

simultaneous sampling of soil and stream water at high frequency is a novelty in DOM studies and to our knowledge, this has never been studied by DOM molecular analysis to investigate the transfer of DOM from soils to rivers during rain events. The in-stream process was also simulated in order to measure its potential impact on the DOM molecular composition during storm events.

## 2 Material and methods

### 2.1 Study site

The study was conducted in the Kervidy-Naizin catchment (Fig. 1), a 4.9 km² headwater catchment located in central Britany, France. It belongs to the French Environmental Research Observatory (ORE) AgrHyS which is the site of a long-term monitoring research program aimed at understanding the impact of agricultural intensification and climate change on hydrologic processes and water quality. The climate is oceanic temperate with an annual mean precipitation of 837 mm and

an annual mean temperature of 11.3°C between 1994 and 2016. Previous studies conducted on this site provide evidence for the structuration of the hydrological year into three different periods: period A where the water table reaches the surface in down-slope wetlands but stays below the surface in the slope; period B where rainfall intensification is responsible for the rise of water table in the slope which creates a hydrological connection between the slope soils, riparian soils and stream (Fig. 2); and period C which is characterized by the return of the water table to deep soil horizons resulting in the progressive drying of

wetland soils (Aubert et al., 2013; Humbert et al., 2015; Lambert et al., 2013; Molenat et al., 2008). With the water table rise during period B, the water flow path geometry in riparian wetlands changes from a vertical to horizontal circulation.

### 2.2 Water and soil sampling

Between 2014 and 2016, three storm events were studied, all situated in period B: two during the 2014–2015 hydrological year on January 14th and February 12th and one during the 2015–2016 hydrological year on January 7th (Fig. 2). The discharge

was recorded every minute by an automatic gauge station located at the outlet of the catchment. The hourly rainfall was monitored at the weather station. The water table level along the slope was monitored every 15 min by piezometers (Fig. 1).



The difference in the altitude of the water table between two piezometers, denoted ΔH, allowed to follow the rise of the water table in the slope during storm events. Stream water samples were collected by an automatic sampler located at the outlet of the catchment. An increase of > 1 L.s$^{-1}$ during 10 min determined the beginning of the sampling. A one litre sample was collected in polypropylene bottles and stored at 4°C. The sampling frequency varied from 30 min to 1 h depending on the

storm event. Soil solutions were collected manually in 1L glass bottles. A pumping system applied to zero-tension lysimeters allowed to collect the soil solution in the soil macroporosity. This device was located in a wetland area, approximately 20 m from the stream and at a depth of 10 cm in the organo-mineral horizon. Each sample corresponds to the combination of three sub-samples collected in three different zero-tension lysimeters implemented at the same depth. The sampling frequency varied between 1 and 3 hours at the beginning of the event. Then, either one or two samples were collected to cover the two or three

days following the storm event. Surface runoff samples were collected manually two or three times per event in glass bottles, in the channels that appeared in the wetland area at the soil surface during storm events. All of these samples collected during storm events were analysed for their DOC and anion concentrations and molecular composition. During the base flow period, daily monitoring was performed by manually sampling 60 mL of stream water at the outlet in polypropylene bottles for the DOC concentration and anion measurements. Soil solutions and outlet stream waters were each sampled over a 15 day periods

for the DOC and anion concentrations and molecular composition. Base flow and storm flow samples were transported to the laboratory to be treated within 24 h after sampling.

Soil was sampled in the riparian area of the sampling transect. Three samples were collected at 0-10 cm, 10-20 cm and 20-30 cm for molecular characterization.

**2.3 In-stream process simulation**

The in-stream process was identified as a potential DOM source during storm flow conditions. It was simulated by shaking 1 g of soil with 1 L ultrapure water at 200 revolutions per minute for 1 h at room temperature, and performed in triplicate to assess the experimental reproducibility. The soil sample was collected in the riparian wetland of the sampling transect, in the organic horizon. It was previously air dried and 2 mm sieved to remove the organic debris. The use of a 1:1000 soil:water ratio was based on the turbidity recorded in stream at the outlet between 2007 and 2010, which rarely exceeded 1 g L$^{-1}$ in storm

flows (Dupas et al., 2015).

**2.4 Sample preparation and chemical analyses**

Daily stream water samples were filtrated at 0.2 µm. The in-stream process simulation, soil solutions and stream water samples were filtrated successively at 0.7 µm (glass fiber filters, Sartorius, Germany) and 0.2 µm (cellulose acetate filters, Sartorius, Germany). A pre-filtration at 0.7 µm was necessary due to a high suspended matter concentration. As 0.2 µm filters are made

of cellulose acetate, they were rinsed with 0.5 L ultrapure water to prevent any release of organic carbon. This volume was previously determined to reach analytical blank values for the DOC measurements. The concentrations were determined using a Shimadzu TOC-5050A total carbon analyser. The precision of the measurements was estimated to be < ± 5 % based on the





repeated analyses on the sample and standard solutions. The chloride, nitrate and sulphate concentrations were measured by ion chromatography. Anion concentration data were not available for event 3. The water and soil samples were freeze-dried for molecular analysis.

**2.5 Molecular analysis**

Approximately 2 mg of lyophilisate and 10 mg of tetramethylammonium hydroxide (TMAH) were introduced in a reactor and placed in a vertical microfurnace pyrolyser PZ-2020D (Frontier Laboratories). To allow the TMAH reaction to occur, pyrolysis was carried out at 400°C during 1 min. The gases produced were injected directly into a GC-2010 (Shimadzu, Japan) equipped with a SLB 5MS capillarity column (60 m, 0.25 mm i.d., 0,25 μm film thickness) with a split mode (between 10 and 15). The temperature of the transfer line was 321°C, and the temperature of the injection port was 310°C. The oven temperature was

initially 50°C (held during 2 min) and rose to 150°C at 15°C/min, and then rose from 150°C to 310°C (held for 14 min) at 3°C/min. Helium was used as a carrier gas with a flow rate of 1.0 mL/min. After separation by GC, the compounds were detected by a QP2010+MS mass spectrometer (Shimadzu, Japan) operating in the full-scan mode for m/z values comprised between 50 and 600. The transfer line was at 280°C and the molecules were ionized by electron impact using an energy of 70 eV, and an ionization source temperature set at 200°C. The molecules were identified by comparing their full-scan mass spectra

with the library provided by the National Institute of Sciences and Technology (NIST) and research articles (Nierop et al., 2005; Nierop and Verstraten, 2004).

Using the appropriate m/z for each compound, the peak areas were integrated and corrected by a mass spectra factor (Table S1 in Supplement) which correspond to the reciprocal of the proportion of the fragment used for the integration and the entire fragmentogram in the NIST library. The identified compounds were classified into three categories: lignins and tannins (LIG),

carbohydrates (CAR) and fatty acids (FA). The proportion of each compound class was calculated by dividing the sum of the areas of the compounds in this class by the sum of the peak areas of all analysed compounds and expressed as a percentage. LIG markers are classified in three main groups: vanillic units (V), syringic units (S) and coumaric units (C). The ratio of coumaric to vanillic units (C/V) was investigated to trace the degradation state of lignin (Kögel, 1986). An increase in this ratio was previously observed during storm events and was attributed to the mobilization of less biodegraded LIG (Dalzell et

al., 2005; Hernes et al., 2008; Jeanneau et al., 2015). The TMAH reaction of CAR was used to analyse the free and terminal monosaccharides and to differentiate the hexoses (C6), deoxyhexoses (deoxyC6) and pentoses (C5) (Fabbri and Helleur, 1999; Grasset et al., 2009). The equilibrium between plant-derived and microbial origins was based on the deoxyC6/C5 ratio (Rumpel and Dignac, 2006). It is also possible to differentiate between plant and microbial biomarkers for FA. FA with a high molecular weight (> C19:0) are from a plant origin while FA with a low molecular weight (< C19:0) come from a microbial origin except

for C16:0 and C18:0 which can be derived from both (Frostegard et al., 1993; Zelles, 1999). The proportion of microbial markers among the analysed compounds ($f_{mic}$) was calculated according to Jeanneau et al., (2015).



## 2.6 Statistical treatments

Principal components analyses (PCA) were performed using XLSTAT (Addinsoft 2013). Three PCAs were carried out on the molecular compositions. The first one was carried out on the LIG markers, using the relative percentage of each molecule. If two molecular markers were correlated or anti-correlated (> 0.9 or < - 0.9 in Pearson's test), the least intense marker was

removed. The molecules retained as variables are identified in Table S1 in Supplement. The second PCA was carried out on the CAR markers, using the relative percentage of each molecule. Due to the occurrence of correlations between the molecules, they were broken down into three classes: C5, deoxyC6 and C6. The third PCA was carried out on the FA markers, using the relative percentage of each molecule. If two molecular markers were correlated or anti-correlated (> 0.9 or < - 0.9 in Pearson's test), the least intense marker was removed. The molecules retained as variables are identified in Table S1 in Supplement. For

these three PCAs, the coordinates of the samples on axis F1, which represents the maximum of variance, were used as a proxy to investigate potential differences in the distribution of the target compounds among these three classes between the soil, surface runoff, stream water and soil solutions during the base flow and storm flow periods. Significant differences were identified using Dunn's multiple comparison test.

Two additional PCAs were carried out to investigate the contribution of the sources (soil solution, surface runoff and

groundwater) to stream water during storm flow conditions. These PCAs were performed using chloride, nitrate and sulphate concentrations that could be considered as tracers due to their contrasting between source concentrations (Christophersen et al., 1990). Since these data were not available for event 3, this statistical treatment was carried out for events 1 and 2. The PCAs were calculated with groundwater, soil solution and surface runoff samples as observations and stream water samples as additional observations. The groundwater concentrations were based on annual samples taken during period B from 2012

to 2014. Since the stream water samples plotted inside the triangle formed by the three end-members, their contributions were calculated by solving a system of equations with three unknowns using their coordinates on axes F1 and F2. The sum of the variance explained by axes F1 and F2 were 82.8 % and 87.3 % for event 1 and 2, respectively. Given that the soil solution end-member was not fixed due to the decrease in the nitrate and chloride concentrations during storm events (Fig. S1 in Supplement), the coordinates of this endmember for the resolution of the system of equation was adapted as a function of the

sampling time.

## 3 Results

### 3.1 Hydrology

Event 1 (Fig. 3a) was characterized by intense precipitation with 43.5 mm of rainfall between the 13th and 16th of January 2015. First, 7 mm of rainfall fell on January 13th which resulted in an increase in the discharge, from 70 to 95 L s$^{-1}$, and ΔH,

from 1.22 to 1.47 m. The discharge returned to the pre-event level but regular rainfall allowed to maintain a high ΔH value. The second precipitation event was larger with steady rainfall during 15 hours for a total amount of 31 mm, and a maximal





intensity of 5 mm h$^{-1}$. This rainfall was responsible for an increase in the discharge, from 89 to 660 L s$^{-1}$, and ΔH, from 1.46 to 1.56 m. A rapid decrease in the discharge occurred at the end of the rainfall. However, ΔH remained higher than 1.45 m two days after the event.

Event 2 (Fig. 3b) was characterized by a smaller total rainfall amount with 18 mm between the 12th and 15th of February 2015, with a maximal intensity of 3 mm h$^{-1}$. Precipitation occurred discontinuously leading to three successive increases in the discharge and ΔH. Over the whole event, the discharge increased from 85 to 175 L s$^{-1}$ and ΔH increased from 1.02 to 1.41 m. The discharge decreased rapidly to 100 L s$^{-1}$, 48 hours after the end of the rainfall. The decrease in ΔH occurred more gradually to reach 1.17 m, 48 hours after the end of the rainfall.

Event 3 (Fig. 3c) was characterized by the establishment of two successive storm flow episodes. The first occurred on the 6th and 7th of January 2016 with 15 mm. Continuous rainfall occurred with a maximal intensity of 3 mm h$^{-1}$ leading to a rapid increase in both the discharge (from 101 to 277 L s$^{-1}$) and ΔH (from 1.29 to 1.53 m). The end of the rainfall induced a rapid decrease in both the discharge and ΔH. The second rainfall episode happened between the 8th and 11th of January 2016 with a total rainfall amount of 30 mm and a maximal rainfall intensity of 3.5 mm/h. These discontinuous rainfalls were responsible for two successive discharge increases which ranged from 112 to 281 L s$^{-1}$. Within 48 h after the end of the rainfall, the levels returned to the pre-event discharge and ΔH levels.

### 3.2 Molecular composition of the SOM

In the SOM, LIG, CAR and FA represented 33 ± 14, 7 ± 1 and 60 ± 13% (mean ± standard deviation; n = 3) of the target compounds, respectively. For CAR, 67 ± 3% of the compounds were hexoses which were mainly derived from cellulose and 13 ± 3% of the FA were derived from cutines and suberines. Among all of the target compounds, 17 ± 6% were from a microbial origin (Fig. 4).

### 3.3 Molecular composition of the soil solution DOM

Over the two hydrological years, the DOC concentration of the soil solution in period B varied from 10.3 to 15.0 mg L$^{-1}$ during base flow conditions (Fig. 3). During the three investigated storm events, rainfall induced modifications of the DOC concentrations in the soil solutions. During event 1, the DOC increased by 2.4 mg L$^{-1}$ and then remained stable. During event 2, the DOC concentration increased by 2.7 mg L$^{-1}$. The beginning of event 3 was characterized by a 2 mg L$^{-1}$ decrease, then the DOC concentration remained 0.6 mg L$^{-1}$ above the base flow concentrations until the end of the sampling period which was characterized by a decrease in the discharge. During event 1, the nitrate concentrations decreased from 7.0 to 0.6 mg L$^{-1}$ whereas the sulphate and chloride levels stayed stable. For event 2, the sulphates stayed stable but the nitrates and chlorides decreased from 23.8 to 8.1 mg L$^{-1}$ and 56.7 to 45.5 mg L$^{-1}$, respectively (Fig. S1 in Supplement).

During the base flow period, LIG, CAR and FA represented 55 ± 11, 10 ± 4 and 35 ± 13% (mean ± standard deviation; n = 13) of the analysed target compounds, respectively. Among all of the samples for the base flow and storm flow soil solutions, FA derived from cutines and suberines were never identified. Compared to the base flow period, events 1 and 3 were





characterized by higher proportions of LIG and CAR, and lower proportions of FA. The proportion of microbial markers ($f_{mic}$) was $36 \pm 12\%$ during the base flow period and decreased to $19 \pm 5\%$ and $24 \pm 8\%$ during events 1 and 3, respectively. For event 2, the LIG, CAR, FA, and $f_{mic}$ values were not significantly different from the base flow conditions (Fig. 4). During base flow, the C/V values in soil solution ranged from 0.1 to 0.3. Storm flow conditions were responsible for an increase in

the C/V ratio with a variable intensity depending on the event. During event 1, the C/V ratio ranged from 0.9 to 2.4 and was six times higher than in the base flow conditions from the first sampling points. During event 2, the C/V ratio ranged from 0.1 to 0.7. It slowly increased with the ΔH values and remained high even after ΔH started to decrease. During event 3, the C/V ratio ranged from 0.1 to 0.3. It slowly increased with the ΔH values but rapidly returned to the base flow level during the decrease in ΔH (Fig. 3).

Based on the PCA analysis, the distribution of the LIG markers is significantly different from the base flow period for event 1. For the FA distribution, events 1 and 3 are significantly different from the base flow distribution. However, the CAR distribution did not vary between the base flow and storm flow distributions. For event 2, the distribution of LIG, CAR and FA are not significantly different from the base flow period when the whole event is considered (Fig. 5). However, when considering the temporal evolution, the three first samples were not significantly different from the LIG distribution during

the base flow conditions, while the following samples were significantly different (Fig. S2 in Supplement).

### 3.4 Molecular composition of the surface runoff

The DOC concentrations in surface runoff ranged from 8.9 to 27.1 mg L$^{-1}$. LIG, CAR and FA represented $58 \pm 5$, $18 \pm 4$ and $24 \pm 8\%$ (mean ± standard deviation; n = 5) of the target compounds respectively and $25 \pm 7\%$ of these compounds were of microbial origin. (Fig. 4). The C/V ratio ranged from 0.5 to 1.8. The LIG distribution corresponded to the distribution observed

for soil solutions, stream water and soil during event 1. The CAR distribution corresponded to the distribution observed for stream water during the base flow and storm flow periods. The FA distribution was intermediate between the distributions observed during the storm flow and base flow periods in soil solutions and stream water (Fig. 5).

### 3.5 Molecular composition of the stream water DOM

Over the two hydrological years, the DOC concentration in stream water varied from 3.5 to 5.9 mg L$^{-1}$ during the base flow

period in period B (Fig. 3). With the establishment of storm flow, the magnitude of the increase in the DOC concentration was event dependent. Events 1 and 3 reached maximum concentrations of 16.1 mg L$^{-1}$ and 15.2 mg L$^{-1}$, respectively. A smaller increase was measured for event 2, which reached a maximum DOC concentration of 10.3 mg L$^{-1}$. The increase in the DOC concentration happened quickly after the increase in discharge, and decreased rapidly during the falling limb of the hydrograph for events 2 and 3, while the falling limb of the hydrograph was not sampled for event 1.

During the base flow period, LIG, CAR and FA represented $44 \pm 9$, $11 \pm 5$ and $45 \pm 12\%$ (mean ± standard deviation, n = 8) of the target compounds, respectively. The microbial biomarkers ($f_{mic}$) represented $46 \pm 12\%$ of the target compounds. Compared to the base flow period, events 1 and 3 were characterized by higher proportions of LIG and lower proportions of



FA (Fig. 4). The target molecules are primarily from a plant origin as indicated by the low $f_{mic}$ value. The relative proportions of CAR were not significantly different from those during the base flow period. For event 2, the proportions of LIG, FA, CAR and the $f_{mic}$ value were not significantly different from the base flow period (Fig. 4). During base flow conditions, the C/V values ranged from 0.13 to 0.20. The magnitude of the increase in the C/V ratio during storm flow conditions was event dependent with maximal values of 0.82 for event 1 and 0.50 for events 2 and 3 (Fig. 3).

Based on the PCA analysis, the distribution of the LIG markers was significantly different from that in base flow conditions for event 1 with a shift toward soil distribution during storm flow (Fig. 5). For the FA distribution, events 1 and 3 were significantly different from the base flow conditions. The distribution of CAR did not vary between the base flow and storm flow conditions (Fig. 5).

## 3.6 Molecular composition of the in-stream process DOM

Samples obtained after 1 h of shaking were characterized by low DOC concentrations (1.1 ± 0.1 mg L$^{-1}$ mean ± standard deviation; n = 3). Among the target molecules, 19 ± 2% were LIG markers, 22 ± 3% were CAR markers and 59 ± 3% were FA markers. The target molecules were mainly from a microbial origin with a $f_{mic}$ value of 59 ± 2%. The lignins produced were characterized by a C/V ratio (0.19 ± 0.01) similar to the soil solutions and stream water sampled during the base flow period.

## 4. Discussion

### 4.1 Is the DOM composition modified in soil solutions during storm events?

The high frequency sampling of soil solutions revealed that the molecular composition of the soil solution was modified during storm events. The establishment of storm event conditions induces a modification in the LIG, CAR and FA proportions as well as the molecular distribution in these classes (Fig. 4 and 5). As indicated by the lower $f_{mic}$ values measured during soil solution storm events compared to the base flow period, soil solution DOM predominantly came from a plant origin (Fig. 4). The increase in the C/V value during storm events revealed that the DOM was composed of lignins that were less biodegraded compared to the base flow DOM (Hedges and Weliky, 1989; Opsahl and Benner, 1995) (Fig. 3).These modifications were recorded during the storm event and during the falling limb of the hydrograph for events 2 and 3. Therefore, this implies that the mechanism responsible for the mobilization of this DOM is persistent after the return to the pre-event discharge levels. Moreover, the intensity of the variations was event dependent. This could be due to the intensity of the mechanism responsible for mobilizing this MOD during flood events.



## 4.2 What are the hydrological drivers of these modifications?

Events 1 and 2 were characterized by different hydrological conditions. The total rainfall amount during event 1 was 43.5 mm with maximal ΔH values that reached 1.57 m. Event 2 was characterized by 18 mm of rainfall and a lower increase of ΔH which reached 1.41 m (Fig. 3). Moreover, the molecular composition of the soil solution DOM sampled during these two storm
events was modified compared to the base flow conditions but with different intensities. The intensity of the molecular composition modification was higher for event 1 compared to event 2 (Fig. 5). As the intensity of storm flow conditions and more particularly the flow rate in soils is known to increase colloidal and particulate mobilization (Kaplan et al., 1993; Majdalani et al., 2008; Zhang et al., 2016), which can be bound with organic matter (Laegdsmand et al., 1999), we hypothesize that hydrological conditions were responsible for this variability between storm events. To test this hypothesis, the distribution
of LIG, which showed more significant molecular modifications, was investigated as a function of ΔH for soil solutions sampled during base flow and storm flow conditions (Fig. 6). These results clearly evidenced the relationship between the rise of the water table in the slope and the intensity of the molecular composition changes. The more intense modifications of the LIG composition were recorded for the highest ΔH values. Therefore, the rise of the water table in the slope seems to control the mechanisms responsible for the mobilization of DOM characterized by different molecular compositions. However, the
LIG composition in soil solutions during event 3 were not significantly different from those in base flow conditions despite high ΔH values (Fig. 5 and 6). These observations highlight that other parameters may be involved in soil solution DOM production during storm events. Events 1 and 3 were characterized by a comparable rainfall amount with 43.5 and 30 mm, respectively, and maximal ΔH values of 1.57 and 1.52 mm, respectively (Fig. 3). However, a first storm event occurred two days before event 3. This pre-event was responsible for the modification of the stream DOM composition as evidenced by the
increase in the C/V value in stream water (Fig. 3). Even if the soil solution had not been sampled, we can hypothesize that this pre-event may have been responsible for the establishment of suitable conditions for DOM mobilization in soil solutions. To explain the lack of variations in the LIG composition found in the soil solution observed when the ΔH values are high, the hypothesis could be that a limited amount of colloids bound with the DOM were available for mobilization (Jarvis et al., 1999). The occurrence of the first pre-event two days before event 3 could have nearly completely depleted the supply of colloids
available for mobilization. Despite the establishment of suitable conditions during event 3, therefore a significant molecular composition could not have been observed.

## 4.3 Are these modifications recorded in the stream DOM?

As evidenced in this study and previous works (Jeanneau et al., 2015), the stream DOM composition was modified during storm events. These modifications are recorded during the event, as well as during the falling limb of the hydrograph when the
level of discharge returned to the base flow conditions for events 2 and 3 (Fig. 3). Three possible origins could explain the modification of the stream DOM composition.





The first one is proposed by Dalzell et al., (2005) and Hernes et al., (2008) who attributed the modifications of the stream DOM composition to the chemical equilibrium between soil particles in the stream. The experimental modelling of this process by shaking soil with water was found to produce small amounts of DOC compared to the increase in the DOC concentration measured over the different storm events. Moreover, the DOM produced was characterized by low C/V values, in contrast

with the high C/V values measured in the stream during storm flow. Consequently, assuming that the experimental conditions were representative of natural conditions, the contribution of this mechanism to DOM production could be considered as negligible in headwater catchments.

The second origin could be the contribution of the soil solution to the stream. Previous works performed on the Kervidy-Naizin catchment (Morel et al., 2009) or in other headwater catchments (Inamdar and Mitchell, 2006; Van Gaelen et al., 2014) have

shown that during base flow periods, the stream water was mostly sustained by deep and shallow groundwater. However, during storm events, the increase in discharge was mostly due to the increase in the soil solution contribution (Lambert et al., 2011; Morel et al., 2009). As the same modifications were recorded in the soil solution and stream DOM composition during storm events, the soil solution may be a possible origin of stream water DOM modification during storm events.

The third origin could be the contribution of surface runoff. During storm events, it may represent a large flux of water (Delpla

et al., 2011) containing a large amount of DOC and POC into the stream (Caverly et al., 2013). Furthermore, surface runoff DOM and soil solutions sampled during storm events have a similar molecular composition (Fig. 4 and 5).

To investigate the contribution of the three sources (soil solution, surface runoff and groundwater) to the stream during storm events, PCAs were performed using the chloride, nitrate and sulphate concentrations. The use of chemical components such as these is common in order to trace the contribution of sources to the stream (Hooper et al., 1990; Lambert et al., 2014; Morel

et al., 2009). The soil solution analysis revealed variable concentrations during the event (Fig. S1 in Supplement). In order to take this variability into account, the proportion of each source used in the PCA analysis was calculated using the soil solution that temporarily corresponds to the stream water. The three sources that contributed to stream discharge during storm flow satisfactorily explain the chemical evolution of the stream water. The simulated DOC concentrations were calculated by multiplying the proportions of each source by their respective DOC concentrations. The models match the observations as the

correlation coefficients between the measured and estimated values are 0.67 and 0.74 for events 1 and 2, respectively (Fig. S3 in Supplement). The estimated fractions were used to calculate their respective contribution to the stream DOC concentration. During events 1 and 2, deep groundwater contributes less than 1 mg L$^{-1}$ to DOC export. Most of the DOC exported by the stream comes from soil solutions and surface runoff (Fig. 7). Events 1 and 2 occurred when the soils were saturated over a large part of the catchment as indicated by the high ΔH value (Fig. 3), which favoured the generation of surface runoff

(Bronstert and Bardossy, 1999). Due to these conditions, the higher rainfall amount during event 1 (31 mm during 15 h) than during event 2 (18 mm during 41 h) could explain the higher surface runoff contribution to the stream DOC. The relative proportion of each of these two sources varied both during the event as well as among events, depending on their hydrological characteristics. Consequently, changes in the DOM molecular composition in soil solution and the contribution of surface runoff to DOM export could be responsible for the modification of the molecular composition observed in the stream water.



## 4.4 Conceptual model for colloidal-DOM mobilisation in soil solutions during storm events

During base flow conditions, DOM came from biotic and abiotic solubilization in the soil microporosity (Toosi et al., 2012). DOM is then transferred to the soil macroporosity by diffusion which is driven by concentration gradients. As a reactive component, DOM can interact with metals and minerals during its transfer along the micro-to-macroporosity continuum. Thus,

DOM can be adsorbed on mineral or clay surfaces (Jardine et al., 1989; Moore et al., 1992) and can be biodegraded by microorganisms according to meeting probabilities (Dungait et al., 2012). Compared to base flow conditions, the increase in the hydrological gradient during storm events induces an increase in the water velocity in the macropores. This increase should not impact the DOM diffusion rate, resulting in a decrease in the DOC concentration in the soil solution due to a dilution effect (Easthouse et al., 1992). The increase in the DOC concentrations and the modification of the composition of the soil DOM

during storm events (Fig. 3, 4 and 5) implies that an additional mechanism of DOM solubilization should be considered. This mechanism would be dependent on the hydrological gradient (Fig. 6) and the pre-event hydrological conditions as illustrated by the comparison between events 1 and 3.

From these observations we can formulate two hypothesis regarding the mobilization of DOM during storm events. First, DOM could come from the mobilization of colloids and soil particles containing organic matter. Numerous studies have

highlighted the mobilization of colloids and soil particles in columns (Laegdsmand et al., 1999; Majdalani et al., 2008; Mohanty et al., 2015; Zhuang et al., 2007) and field studies (Jarvis et al., 1999; Zhang et al., 2016) due to an increasing water velocity. The rise of the water table during storm flow conditions induced an increase in the water pressure and velocity in the soil macroporosity, which could be related to a piston-like effect (Zhao et al., 2017). This would lead to an application of shear forces on the colloids and particles located on the walls of the macropores (Bergendahl and Grasso, 2003; Shang et al., 2008).

If the shear forces are stronger than the forces that attach the colloids to the macropore wall, colloids will be released into the soil solution (DeNovio et al., 2004; Ryan et al., 1998) (Fig. 8). This mechanism of colloidal and particulate destabilization is consistent with the threshold highlighted in Fig. 6 where it seems that ΔH must be exceeded in order to create a sufficient shear force to initiate the destabilization. This is also consistent with the largest modification of the LIG composition recorded for the highest ΔH since these hydrological conditions are responsible for the largest colloidal destabilization. Moreover, the pre-

event hydrological conditions are also consistent with this physical destabilization. The colloidal and particulate supply available for mobilization appears to be size-limited and renewable (Jarvis et al., 1999; Majdalani et al., 2008). Thus, the pre-event hydrological conditions will impact the possibility to rebuild this supply.

However, the chemical composition of DOM during storm events differs from the SOM composition. The molecular analysis of soil DOM from the three storm events investigated highlight the differences in the molecular composition variations that

exist between LIG, FA and CAR. For the event 1, where the ΔH values were the highest, the distribution of LIG in the dissolved phases was similar to their distribution in SOM (Fig. 5). However, CAR and FA were characterized by a different evolution. Variations in the FA and CAR distributions between base flow and storm flow conditions are low. Depending on the events, the FA distribution is significantly different from the base flow conditions, and the CAR distribution was not significantly



different from the base flow conditions for the three events. Furthermore, the distribution of both FA and CAR in DOM always remains significantly different from their distribution in SOM (Fig. 5). Which mechanism could explain the different behaviour of these three molecular classes? Among all of the FA identified in SOM, $13 \pm 3\%$ were FA derived from cutines and suberines which came from a plant origin (Chefetz et al., 2002; Nierop and Verstraten, 2004). These molecules were not identified in

the soil solutions. This absence could be linked to their high hydrophobicity (Kolattukudy, 1984). Similarly, $67 \pm 3\%$ of the CAR identified in the soils are hexoses, mainly coming from the thermochemolysis of cellulose, a polymer of glucose which is highly hydrophobic (Krässig, 1993). There is very little cellulose in the solution which could explain the differences in the CAR distribution between the soil and soil solutions. However, LIG are less hydrophobic than FA and CAR. These different behaviours of macromolecules during their solubilization from the soil to the soil solution are thus consistent with the

hypothesis of a combined physical mechanism and chemical segregation based on the hydrophobicity of the macromolecules. This chemical segregation could take place during the formation of colloids and particles on macropore walls or upon their mobilization. Since a comparable composition was observed between the soil solution and the surface runoff DOM, the same mechanism could be applied for surface runoff with shear forces applied by the runoff of water on the soil surface.

## 5. Conclusion

For the first time, the molecular composition of DOM was simultaneously investigated in soil solutions, surface runoff and stream water during storm events with high frequency sampling. The major conclusions of this study are the following:

(i) The modifications of the DOM composition in soil solutions and the generation of surface runoff are responsible for the changes in the DOM composition in stream water during the establishment of storm flow conditions.

(ii) The changes in the DOM molecular composition is due to a combination of physical and chemical mechanisms.

The increase in the water velocity in the macropores induces the destabilization of colloids and soil particles composed of organic matter. A chemical segregation could be responsible for the changes in the molecular composition between SOM and soil DOM based on the hydrophobicity of the organic macromolecules.

(iii) Low water velocity and favourable hydrological conditions in soils are necessary to rebuild the colloidal and particulate supply. Therefore, their mobilization during storm events are dependent on the pre-event hydrological

conditions.

These changes in the DOM composition should be taken into account for a better understanding of micropollutant mobility. As the complexation of micropollutants (e.g. pesticides) with OM is mainly driven by hydrophobicity, the export of less biodegraded DOM during storm events may have increased their diffusion across the environment. Moreover, an increase in storm frequency and intensity over the next decades, as predicted by climatologists (Coumou and Rahmstorf, 2012), could

increase the export of DOM produced during storm events and thus the dispersion of pollutants in the environment.



## Acknowledgements

This study used equipment from and data collected for the AgrHyS environmental research observatory (http://www6.inra.fr/ore_agrhys). We thank Sen Gu and technical staff from INRA and Géosciences Rennes for their assistance during the field sampling. We also thank all of the people involved in the maintenance of the field installations and the creation

of the databases. Dr. S. Mullin post-edited the English style (www.trad8.eu/us/sara-mullin.html)

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



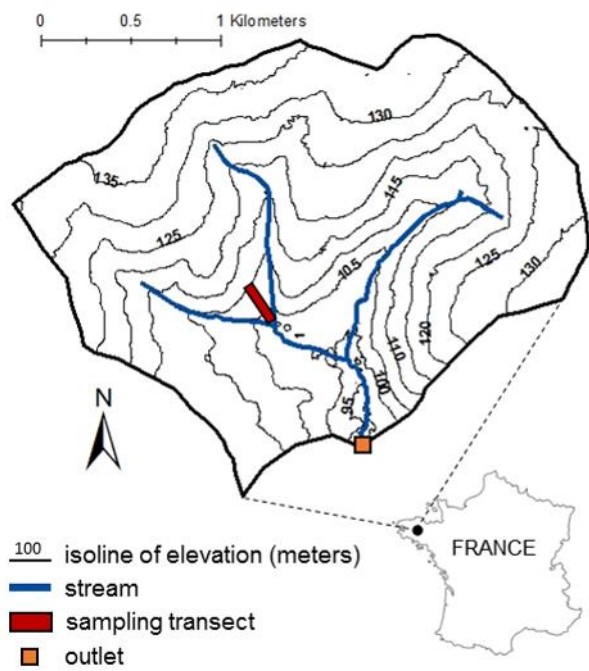

**Figure 1: Map of the Kervidy-Naizin catchment (Britany, France). The soil solutions were sampled in the wetland area of the transect. The stream waters were sampled at the outlet of the catchment.**





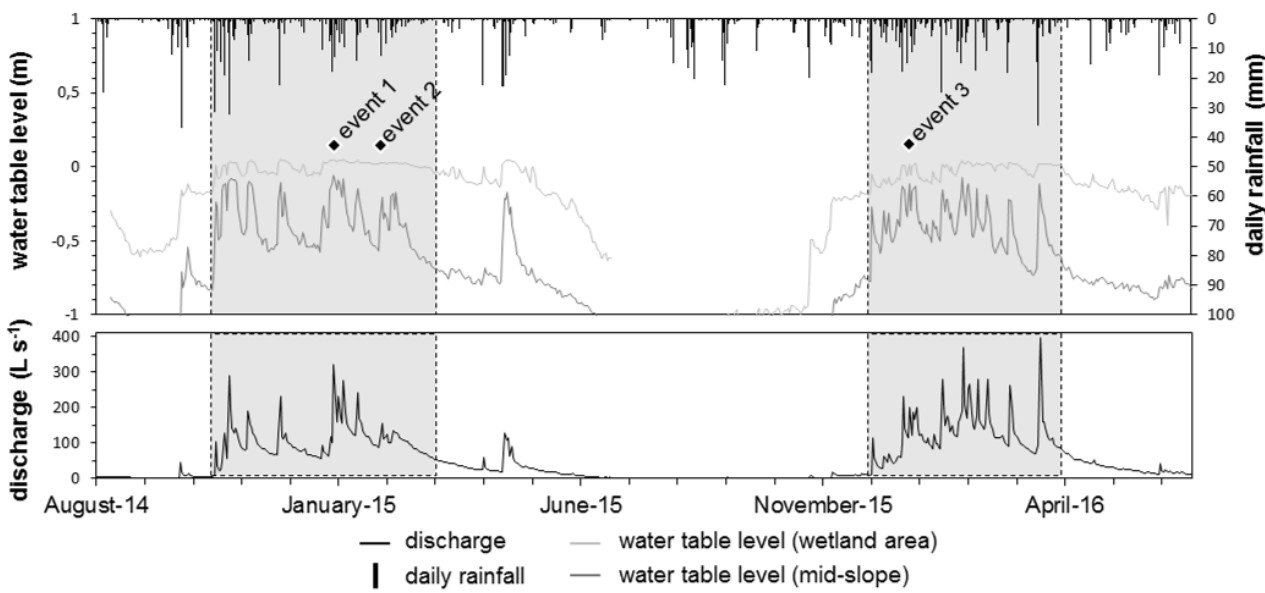

5  **Figure 2: Daily rainfall, water table level and discharge during hydrological years 2014/2015 and 2015/2016. The storm events were sampled when water table remains in surface horizons in the wetland area (hydrological period B – grey areas). This period is characterised by hydrological connectivity between mid-slope and wetland soils during rain events.**





**Figure 3: Hourly rainfall, discharge and ΔH for events 1, 2 and 3. Evolution of DOC concentration and C/V in stream water and soil solution.**





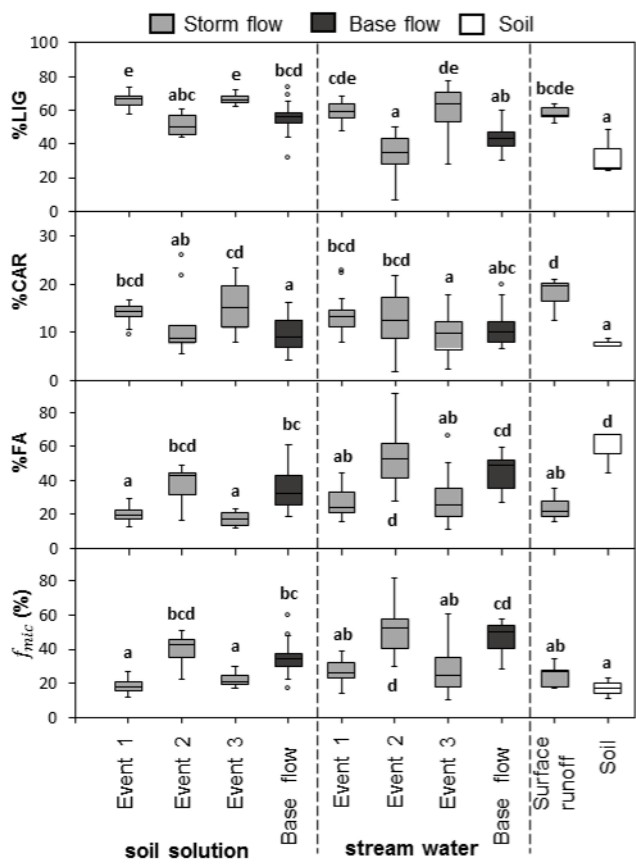

**Figure 4: Box-plot of %LIG, %CAR, %FA and $f_{mic}$ in soil, surface runoff, stream water and soil solution during storm-flow and base-flow conditions. Letters not shared across box plots indicate significant mean differences using Dunn's test of multiple comparison.**





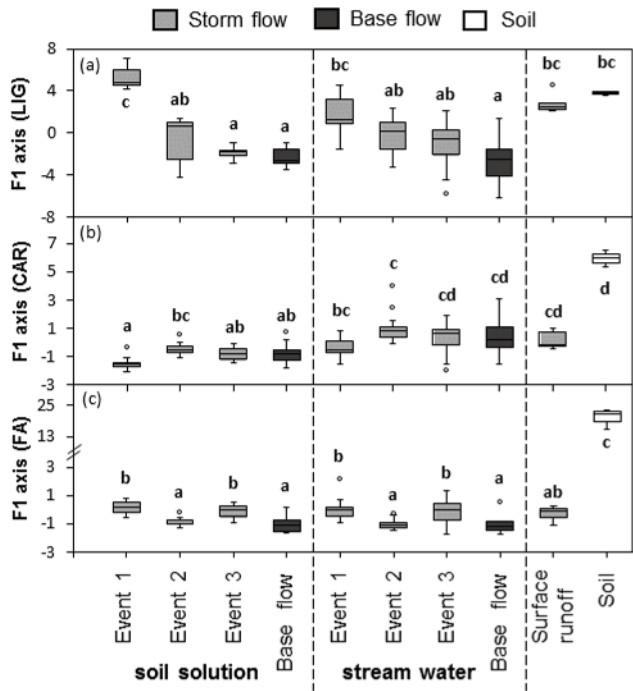

**Figure 5: Coordinate on F1 axis from (a) LIG, (b) CAR and (c) FA molecular distribution PCA analysis for soil, surface runoff, stream water and soil solution during storm-flow and base-flow. Letters not shared across box plots indicate significant mean differences using Dunn's test of multiple comparison.**





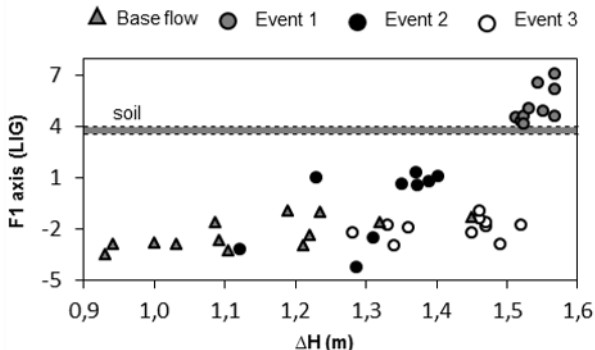

**Figure 6: Evolution of LIG distribution evidenced by PCA analysis as a function of ΔH.**



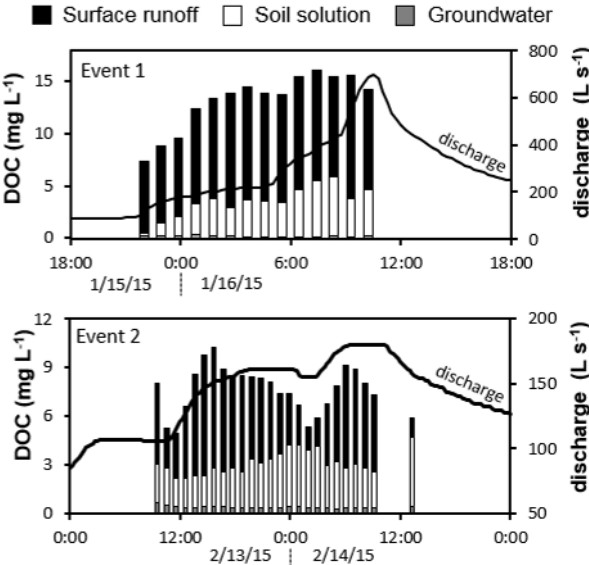

**Figure 7: Estimated contribution of surface runoff, soil solution and groundwater to stream DOC export.**



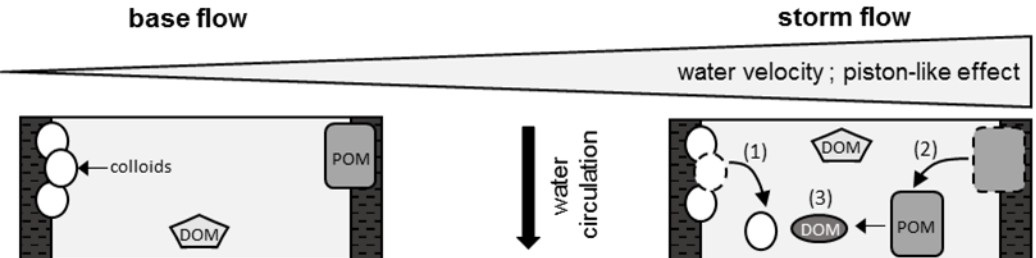

5   **Figure 8: Schematic representation of DOM mobilisation mechanisms involved during storm events in soils. Mobilisation of DOM during storm flow conditions by (1) colloidal destabilisation and / or (2) particulate destabilisation combined with (3) chemical equilibrium processes.**