# Peer review of "New molecular evidence for surface and sub-surface soil erosion controls on the composition of stream DOM during storm events"

_Biogeosciences, 2017_

## Referee Comment (RC1) · P. J. Hernes (Referee) · 11 Aug 2017

P. J. Hernes (Referee)

pjhernes@ucdavis.edu

Linking catchment sources and processes to streamwater chemistry is essential to realizing the full potential of streams and rivers as monitors for changing terrestrial environments. I have spent a lot of time on this topic, and it is not an easy nut to crack. It is great to see others working on this as well, as we need a critical mass of scientists looking at it from different angles. This paper focuses on sources of increased DOM in streams during storm events and presents a conceptual model for mobilization of colloids and particles within the soils and subsequent release of DOM. Based on experiments that we have done in our lab, I agree that simple desorption of OM from

soils is not a strong enough source to account for increased DOM concentrations in streams, and so it is a bit unclear as to how the authors think that the conceptual model in Fig. 8 is fundamentally different from desorption of OM from soils in terms of generating enough to account for the increase? Clearly, there is a mass balance issue that needs to be factored in – it's not just about matching chemical composition, there has to be enough DOM from the various sources as well. At the risk of appearing self-serving, I would encourage the authors to have a look at the recently published Hernes et al. (2017) in Frontiers in Earth Sciences, which also focuses on rain events, also has subsurface flow/sampling, also involves lignin chemistry, and argues that simple litter/duff leaching (or leaching during stemflow and canopy throughfall) can produce all the DOM needed to account for increases in stream DOM, and that perhaps the main role of the soils is simply to modify and reduce the pulse of DOM flowing through to the streams. Granted, there are significant differences between the two systems, and the soil desorption experiments demonstrate that it does not have to be mechanistically either/or, but at a minimum, the authors need to address the conflict between Fig. 8 and the soil desorption results.

Regarding the sourcing, the way in which the pyrolysis target compounds have been normalized to the total response of all target compounds introduces some fundamental problems in the data analysis in that any change in one component percentage necessitates the opposite change in at least one other component. It's a zero sum game, and not always straightforward on how to interpret those changes – does an increase in the %LIG mean that more lignin was produced, or does it simply mean that carbohydrates or fatty acids were degraded or produced less? There are at least two interpretations for every change in a single component, and you have to be extremely careful to sort out which is which. Rather than interpreting the percentages straight up, it might be beneficial to multiply the percentages times the DOM concentrations and makes some plots of those concentrations with time to better evaluate what is increasing vs. what might be decreasing or degrading. (Of course, there is also an inherent assumption that the yield efficiency of your target compounds is constant across sample types and

concentrations.) Again, the conceptual problem that I wrestled with in Hernes et al. was the capacity of the soils to modify chemical composition during lateral flow through the soils. The mass balance says that there has to be significant sorption or degradation (or both) happening, and that almost any of the plant litters/duffs was producing enough lignin to account for streamwater chemistry.

There may be some interpretive value in considering the DOC hysteresis of the flow events: For any given discharge, was the DOC higher on the rising or the falling limb? If it's higher on the falling limb, this could indicate a lag time in whatever processes are at play in mobilizing new DOM from the litters or soils. If it's higher on the rising limb, this suggests that the source of DOM was already mobilized and perched in the soils, waiting for a flush.

Figures 4 and 5 are not all that helpful, in my opinion, as there is too much going on and they are hard to interpret. I don't know what the random circles are – data points that are being arbitrarily excluded from the statistics? If so, why? It's confusing how the Event 2 %LIG in Fig. 4 can have the "a" label in common with the soil – they barely look like they overlap. Baseflow %LIG surely looks like it should overlap with "e", especially if the extra data points are factored in. There are numerous examples of confusing similarities or differences within these two figures. Also conspicuously missing from these plots are any indication of the number of samples/datapoints per box. Statistics are merely an interpretive tool, but they can also be very misleading at low n, or when outliers are being excluded, or countless other factors when running regressions, so you want to include the information necessary so that we can evaluate whether the statistics are meaningful or meaningless.

It is critical to keep in mind phase changes and the potential for fractionation when comparing solids to dissolved. The trends in Opsahl and Benner (1995), for example, may not all be relevant to the dissolved phase.

The delta-H term and water table levels were confusing to me. Delta-H is supposedly

the difference between the two piezometers, and yet when I look at Fig. 2, for example, the difference between the water tables seems to be 0.1 to 0.5 m most of the time. Yet delta-H is presented as 1.0 to >1.5 m. I am obviously missing something. What is the 0 reference point for the water table in Fig. 2?

―――――――――――――――――――

---

## Referee Comment (RC2) · Anonymous Referee #2 · 16 Aug 2017

Dear,

I have finished my review of the manuscript "New molecular evidence for surface and subsurface soil erosion controls on the composition of stream DOM during storm events", submitted to Biogeosciences by Marie Denis and co-authors.

I find that the paper is written in a clear way, the authors have targeted three clear research questions using an appropriately chosen dataset and methodology. The line of thought is clear. The method of analysis chosen (thermally assisted hydrolysis and methylation) is appropriate to target the variety of compounds that allow to characterize DOM in soils and headwater. In my opinion simulating the in-stream process does not

add much to the manuscript.

Minor comments:

P2 L 19. Give some references for these mixing analysis and isotopic studies.

P4 L 8. Could the authors comment on the possible effect on the soil/water equilibrium and the associated liberated DOC that is caused by deploying a zero-tension lysometer?

P4 L 15. Why was the soil only sampled in the riparian transect? Is there no difference expected with the slope soils in molecular composition?

P5 L23-25. This sentence does not really belong in the methods section.

P5 L31. This should be Jeanneau et al. (2014). "Jeanneau, L., Jaffrezic, A., Pierson-Wickmann, A.-C., Gruau, G., Lambert, T., and Petitjean, P.: Constraints on the Sources and Production Mechanisms of Dissolved Organic Matter in Soils from Molecular Biomarkers, Vadose Zone J., 13, 2014"

P5 L29. The method used here is developed for soils (Jeanneau et al., 2014). How are you taking phytoplankton fatty acids into account? These are generally polyunsaturated long-chain compounds, did you find any of these? Would it be beneficial to use ratios between specific biomarkers for plants, bacteria, possibly fungi and phytoplankton biomass?

P6 L20. I'm surprised that it is possible to use the values on the PC axes to calculate the relative contribution of the sources. Could you provide a reference that supports this approach?

P6 L20. Why are the axes called F1 and F2 istead of PC1 and PC2 (principal component). That would make the figures more intuitive to read as well.

P6 L24. The coordinates 'were'.

P8 L10. How do you determine that an event is significantly different based on a PCA? Did you use a statistical test? How much of the variance is explained by the first two principal components? This determines how well the ordination space reflects the complete variance between the compounds, and how reliable a statement such as 'significantly different' is.

---

## Author Comment (AC1) · 12 Sep 2017

Thank you for your suggestions and comments to improve this manuscript. Your remarks will be considered to the next version of the manuscript. You will find below a point by point response to your comments reported into square brackets.

[Based on experiments that we have done in our lab, I agree that simple desorption of OM from soils is not a strong enough source to account for increased DOM concentrations in streams, and so it is a bit unclear as to how the authors think that the conceptual model in Fig. 8 is fundamentally different from desorption of OM from soils in terms of generating enough to account for the increase? Clearly, there is a mass balance issue

that needs to be factored in – it's not just about matching chemical composition, there has to be enough DOM from the various sources as well. At the risk of appearing self-serving, I would encourage the authors to have a look at the recently published Hernes et al. (2017) in Frontiers in Earth Sciences, which also focuses on rain events, also has subsurface flow/sampling, also involves lignin chemistry, and argues that simple litter/duff leaching (or leaching during stemflow and canopy throughfall) can produce all the DOM needed to account for increases in stream DOM, and that perhaps the main role of the soils is simply to modify and reduce the pulse of DOM flowing through to the streams. Granted, there are significant differences between the two systems, and the soil desorption experiments demonstrate that it does not have to be mechanistically either/or, but at a minimum, the authors need to address the conflict between Fig. 8 and the soil desorption results].

The analogical modelling of the in-stream process simulated by shaking 1g of soil with 1 L of water was conducted in order to assess the potential contribution of the so called "in-stream process" which was proposed to explain the changes of molecular composition in stream water. This analogical modelling is fundamentally different from the conceptual model proposed in Figure 8 as it takes place in soil macroporosity. These two processes are fundamentally different because of the difference in the soil/water ratio observed in these two location of the soil/stream continuum. The soil/water ratio observed in saturated soils is 1/1.5 when the soil/water ratio used in the in-stream process simulation is 1/1000. This difference in soil/water ratio will therefore impact the solubilization of organic molecules. Following the results of in-stream process simulation we concluded that this mechanism could not explain the changes of molecular composition and the increase of DOC concentration during storm events. However, the determination of relative sources contribution demonstrated that combination of surface runoff (erosion of soil surface and litter leaching) and subsurface erosion (Fig. 8) can explain the increase of DOC observed in the river during storm events.

[Regarding the sourcing, the way in which the pyrolysis target compounds have been

normalized to the total response of all target compounds introduces some fundamental problems in the data analysis in that any change in one component percentage necessitates the opposite change in at least one other component. It's a zero sum game, and not always straightforward on how to interpret those changes – does an increase in the %LIG mean that more lignin was produced, or does it simply mean that carbohydrates or fatty acids were degraded or produced less? There are at least two interpretations for every change in a single component, and you have to be extremely careful to sort out which is which. Rather than interpreting the percentages straight up, it might be beneficial to multiply the percentages times the DOM concentrations and makes some plots of those concentrations with time to better evaluate what is increasing vs. what might be decreasing or degrading. (Of course, there is also an inherent assumption that the yield efficiency of your target compounds is constant across sample types and concentrations.)]

I fully agree about the precautions which must be taken with this type of results, especially when they are used to discuss about biogeochemical functioning. During the discussion of our manuscript, all these results of %LIG, %CAR and %FA are not interpreted in terms of biogeochemical processes. These results are only used to present the differences of DOM composition during the two hydrological conditions (storm event / base flow) and between the different compartments (soil solutions / stream water / surface runoff). The different way you proposed to process the data were used for the interpretation of datas presented in the first submission of Jeanneau et al. (2014) and was highly criticized by the reviewers. For these reasons we chose not to go further for in the interpretation of the data, and to present the data as percentages.

[Again, the conceptual problem that I wrestled with in Hernes et al. was the capacity of the soils to modify chemical composition during lateral flow through the soils. The mass balance says that there has to be significant sorption or degradation (or both) happening, and that almost any of the plant litters/duffs was producing enough lignin to account for streamwater chemistry.]
The three storm events we followed in this study occurred when soils are completely saturated, even during base-flow period. Therefore, water circulation in these soils is only horizontal circulation, so we cannot hypothesize vertical circulation of litter leachate to explain the modification of soil solution molecular composition. This precision may be added in the discussion section. However, in the riparian area of the transect, litter leachate will contribute to the increase of DOC concentration in the stream as surface runoff is one of the three sources considered in this study.

[There may be some interpretive value in considering the DOC hysteresis of the flow events: For any given discharge, was the DOC higher on the rising or the falling limb? If it's higher on the falling limb, this could indicate a lag time in whatever processes are at play in mobilizing new DOM from the litters or soils. If it's higher on the rising limb, this suggests that the source of DOM was already mobilized and perched in the soils, waiting for a flush.]

Lambert et al. (2014) and Morel et al. (2009) reported 10 different storm events sampled on the Kervidy-Naizin catchment for which the DOC concentrations in stream waters were higher during the falling limb of the hydrograph. Moreover, 6 storm events which were not added to this study were sampled on the Kervidy-Naizin catchment. 5 of them are also characterized by higher DOC concentrations during the falling limb of the hydrograph. For these 6 storm events, temporal evolution of DOC and discharge are available in the document in supplement. The first event of this study was only sampled during the rising limb of the hydrograph. Consequently it is not possible to see a hysteresis. The second event is characterized by a different evolution with higher DOC concentrations in the rising limb of the hydrograph. The third event is characterized by a small hysteresis where higher DOC concentrations were observed on the falling limb of the hydrograph. For these 3 storm events, temporal evolution of DOC and discharge are available in the document in supplement. The DOC exported at the outlet during storm events came from the surface runoff and the soil solution present in soil macroporosity. Considering the size of the Kervidy-Naizin catchment ($4.9 \ km^2$), a storm event

characterized by intense rains at the beginning of the event could be responsible of the rapid generation of surface runoff, especially during winter when soils in the riparian area are still saturated. These conditions could induce higher concentrations on the rising limb of the hydrograph. For smaller storm events with lower rain amounts, soil solutions are responsible for a larger part of DOC increase. As transfer of soil solutions to river is a process which occurred more slowly than surface runoff generation, this could be in favor of the observation of higher DOC concentrations in the falling limb of the hydrograph. Consequently, depending of the storm event, the proportion of these two sources changes and could explain why some events are characterized by higher DOC concentrations in the rising limb of the hydrograph while some events are characterized by higher DOC concentrations in the falling limb of the hydrograph.

[Figures 4 and 5 are not all that helpful, in my opinion, as there is too much going on and they are hard to interpret. I don't know what the random circles are – data points that are being arbitrarily excluded from the statistics? If so, why? It's confusing how the Event 2 %LIG in Fig. 4 can have the "a" label in common with the soil – they barely look like they overlap. Baseflow %LIG surely looks like it should overlap with "e", especially if the extra data points are factored in. There are numerous examples of confusing similarities or differences within these two figures. Also conspicuously missing from these plots are any indication of the number of samples/datapoints per box. Statistics are merely an interpretive tool, but they can also be very misleading at low n, or when outliers are being excluded, or countless other factors when running regressions, so you want to include the information necessary so that we can evaluate whether the statistics are meaningful or meaningless.]

In the box-plot representation used in Fig. 4 and 5, the white circles represent the values considered as outliers by the statistical software. However, these values were not excluded from the datasets to perform the statistical tests. The statistical test used to determine the statistical differences is the Dunn's test, a non-parametric test. We chose to use this test because some of the series of data do not follow a normal

distribution. The number of samples per box will be added to the Figures 4 and 5. For soil solutions, there are 10, 9, 10 and 13 samples during events 1, 2, 3 and base-flow respectively. For stream water, there are 14, 25, 18 and 8 samples during events 1, 2, 3 and base-flow respectively. Surface runoff is composed of 5 samples and soil of 3 samples. As we used a non-parametric test for datasets with variable number of samples, this could explain some apparent confusions you listed.

[The delta-H term and water table levels were confusing to me. Delta-H is supposedly the difference between the two piezometers, and yet when I look at Fig. 2, for example, the difference between the water tables seems to be 0.1 to 0.5 m most of the time. Yet delta-H is presented as 1.0 to >1.5 m. I am obviously missing something. What is the 0 reference point for the water table in Fig. 2?]

In Fig. 2, water table level is measured by taking the 0 reference point as soil surface. Consequently, the water table level does not take into account the altitude (above sea level) of the soil surface for the two piezometers implemented along the soil profile. In the contrary, the $\Delta H$ value take into account the difference of altitude between the wetland piezometer and the mid-slope piezometer. Therefore the 0 reference point to calculate the $\Delta H$ is the see level. The 0 reference for the water table level and $\Delta H$ can be specified in the material and method section (Page 3 Line 31) and "water table level" can be changed into "water table depth" in order to clarify this point.

Please also note the supplement to this comment:
https://www.biogeosciences-discuss.net/bg-2017-252/bg-2017-252-AC1-supplement.pdf

**Supplement:**

Variation in stream discharge and DOC concentration during the three storm event followed in this study.

[Figure]

Variation in stream discharge and DOC concentration during six storm events sampled during hydrological year 2013/2014. These storm events ware not added to this study.

---

## Author Comment (AC2) · 12 Sep 2017

Thank you for your suggestions and comments to improve this manuscript. Your remarks will be considered to the next version of the manuscript. You will find below a point by point response to your comments reported into square brackets.

[In my opinion simulating the in-stream process does not add much to the manuscript]

Even if the results from the simulation of the in-stream process does not provide the principal information of this study, we think that these results could help to argue for the transition between the hypothesis of in-stream process previously proposed to explain

the modification of DOM molecular composition in stream water, and our hypothesis who involved surface and sub-surface soil erosion. Indeed, even if the previously proposed in-stream process cannot fully explained the changes of DOM molecular composition observed, it could contribute in part to the modifications observed in stream water. The simulation carried out by agitation of soil and water at an appropriate ratio have shown that the amount of DOC produced is negligible. This result thus allowed to exclude this process to explain the changes of molecular composition observed in stream water.

[P2 L 19. Give some references for these mixing analysis and isotopic studies.]

These references will be added to the text: Bazemore et al. (1994), Klaus and McDonnell (2013), Lambert et al. (2014).

* Bazemore, D.E., Eshleman, K.N., Hollenbeck, K.J., The role of soil water in storm-flow generation in a forested headwater catchment: synthesis of natural tracer and hydrometric evidence, Journal of Hydrology, 162, 47-75, 1994

* Klaus, J., McDonnell, J.J., Hydrograph separation using stable isotopes: review and evaluation, Journal of Hydrology, 505, 47-64, 2013

[P4 L 8. Could the authors comment on the possible effect on the soil/water equilibrium and the associated liberated DOC that is caused by deploying a zero-tension lysimeter?]

Indeed, deploying a zero-tension lysimeter could have possible effect on soil/water equilibrium and therefore on DOC solubilization due to mechanistic soil perturbation. On the Kervidy-Naizin catchment, lysimeters were implemented for long-term sampling experimentation during the summer of 2013. The dataset used in this study is composed of samples sampled in these lysimeters between October 2014 and June 2016. Therefore we could consider that the time laps between the summer of 2013 and October 2014 has been enough to allow the restructuration of the soil around the

zero-tension lysimeters.

[P4 L 15. Why was the soil only sampled in the riparian transect? Is there no difference expected with the slope soils in molecular composition?]

The DOM of soil solutions and stream water were compared to SOM of soils collected at different depth, only in the riparian area of the transect. We chose to compare the molecular composition of DOM with SOM from the riparian area of the transect because all the soil solutions were sampled in this riparian area. As vegetation, hydrology and soil characteristics are different between the riparian area and the slope, we can indeed expect changes in soil molecular composition.

[P5 L23-25. This sentence does not really belong in the methods section.]

This sentence will be modified in order to better fit with the methods section.

[P5 L31. This should be Jeanneau et al. (2014). "Jeanneau, L., Jaffrezic, A., Pierson- Wickmann, A.-C., Gruau, G., Lambert, T., and Petitjean, P.: Constraints on the Sources and Production Mechanisms of Dissolved Organic Matter in Soils from Molecular Biomarkers, Vadose Zone J., 13, 2014"]

There is in fact a reference error. This reference will be modified in the text and added in the bibliography.

[P5 L29. The method used here is developed for soils (Jeanneau et al., 2014). How are you taking phytoplankton fatty acids into account? These are generally polyunsaturated long-chain compounds, did you find any of these? Would it be beneficial to use ratios between specific biomarkers for plants, bacteria, possibly fungi and phytoplankton biomass?]

The method used here was applied for soils but also for stream water and soil DOM by Jeanneau et al. (2014), and could therefore be applied to our dataset. Among all the fatty acids identified using THM-GC-MS, no polyunsaturated long-chain fatty acids were identified in our samples, which does not mean that they are not present

in our samples. These fatty acids could be present but undetectable because they are present in concentrations below the limit of detection. The use of ratios between specific fatty acids biomarkers could be used to assess the impact of the flood events on the microbial activity. However, other methods like PLFA analysis are more suitable than THM-GC-MS to identify the specific fatty acids biomarkers.

[P6 L20. I'm surprised that it is possible to use the values on the PC axes to calculate the relative contribution of the sources. Could you provide a reference that supports this approach?]

We choose to calculate the relative contribution of the sources using PCA to represent the maximum of variance from the three variables (chlorides, nitrates and sulphates) on a single 2D plot. Using the two principal components allow to represents 87.3% of the variance, with 58.3% and 29.0% for PC1 and PC2 respectively. The DOC concentrations estimated from these contributions fits with the DOC concentrations measured in stream water. Even if this method is not the most widely used, these results seems to confirm that this is an appropriate method to determine the relative contribution of the sources.

[P6 L20. Why are the axes called F1 and F2 instead of PC1 and PC2 (principal component). That would make the figures more intuitive to read as well.]

The names of the axis will be changed into PC1 and PC2 in the text and on the figures.

[P6 L24. The coordinates 'were'.]

This sentence will be modified.

[P8 L10. How do you determine that an event is significantly different based on a PCA? Did you use a statistical test? How much of the variance is explained by the first two principal components? This determines how well the ordination space reflects the complete variance between the compounds, and how reliable a statement such as 'significantly different' is.]

The differences between two events were determined using Dunn's test. This statistical test is applied on the coordinate on PC1 axis from the PCA. The results of these tests are given in Figures 5 by the letters added on the top of the box-plots. If two box-plots do not share the same letter, this indicates significant differences between the two datasets. The percentage of the variance explained by the PC1 is 37.4% for LIG, 62.0% for CAR and 39.0% for FA. These informations will be added on Figure 5.

---

## Author Response (AR1)

Deer Marcel van der Meer,

Thank you for your comments and suggestions to improve this manuscript. You will find below a point by point response to your comments with a list of all the changes made in the manuscript, and a marked-up manuscript version. (The page and line numbering corresponds to the marked-up version of the manuscript)

"There are a few things I would like to stress though, I completely agree with P.J. Hernes in that relative contributions as in figure 4 are confusing. I do see that similar distributions probably have similar sources, but increases and decreases are problematic and you need some go from relative changes to absolute changes to get a grip on increasing and decreasing contributions from different sources. Although I understand that in previous cases reviewers have indicated not to do this, I would like to ask you to somehow include information on absolute changes between the different scenarios."

Transformation of relative distribution of %LIG, %FA and %CAR in absolute distribution by multiplying these proportions by the DOC concentration of the sample could not be performed on these results. In fact, it is well established that DOC is not only composed by lignins, fatty acids and carbohydrates. DOC also includes proteins and small organic acids that are not analyzed by THM-GC-MS in this study. Moreover, analysis efficiencies are not the same for the three molecular classes analyzed. For example THM-GC-MS allow to identify only free and terminal monosaccharides from the polysaccharide.

For the parameter $f_{mic}$, we performed the transformation from relative contribution to absolute contribution. The results presented below highlight that most of the DOC increase in stream water during storm event come from a plant origin. In soil solutions, the contribution of microbial DOC and plant DOC during storms 1, 2 and 3 remain quite stable during all the event. However, we don't believe that these results bring new insights and help for the construction of the discussion.

As you and P.J. Hernes underline it, the Figure 4 appears confusing and all the results presented in this figure are not used for discussion but were initially added to the article for sample description. Therefore, we choose to delete these results description in the different sections.

(page 5 lines 23-24 ; page 7 lines 23-24 and 25-27 ; page 8 lines 6-11 and 25-27 ; page 9 lines 8-13 and 24-25 ; page 10 lines 4-7)

Regarding the figure 5, I understand that the choice we made by representing the data using box-plot could appear confusing. Uncertainties represented in the box-plot are due to the evolution of molecular composition during the event since each box-plot contain samples sampled at the beginning, during and at the end of the event.

An alternative option to represent these data would have been to represent the evolution on PC1 axis for the 3 events, for the 3 classes of molecules and for soils solutions and stream water, which represent 18 graphics. This could bring information about temporal variations however this figure would be hard to understand because of the high quantity of information represented on each of the 18 graphics.

Taking into count all these information, I suggest that we could keep the figure 5 as it is but to offer to the reader the possibility to refer to the 18 graphics in supplementary information. Modifications have been performed in this sense.

(page 9 lines 19-20)

[Figure]

"Reviewer two asked some question on the statistical analysis specifically about how to turn values on the PC axes into relative contribution of sources, it would be good if you could elaborate a bit more on how this might work and possible a reference."

The "2.6 statistical treatments" section has been modified accordingly to these remarks. More detailed information has been added about how to proceed the statistical treatment. Two references have also been added to justify the use of Principal Component Analysis in order to use a 2D projection of the data that will represent a maximum of variance (page 6 line 21-32)

**Modifications performed accordingly to P.J. Hernes suggestions**:

"At the risk of appearing self-serving, I would encourage the authors to have a look at the recently published Hernes et al. (2017) in Frontiers in Earth Sciences, which also focuses on rain events, also has subsurface flow/sampling, also involves lignin chemistry, and argues that simple litter/duff leaching (or leaching during stemflow and canopy throughfall) can produce all the DOM needed to account for increases in stream DOM"

Considerations about the possible contribution of litter leachate to the modification of DOM molecular composition and concentration in stream during storm events have been added to the discussion. (page 11 line 26-33)

"The delta-H term and water table levels were confusing to me. Delta-H is supposedly the difference between the two piezometers, and yet when I look at Fig. 2, for example, the difference between the water tables seems to be 0.1 to 0.5 m most of the time. Yet delta-H is presented as 1.0 to >1.5 m. I am obviously missing something. What is the 0 reference point for the water table in Fig. 2?"

The section "2.2 water and soil sampling" has been modified to clarify the difference between the water table level and the $\Delta H$ values. (page 4 line 1)

**Modifications performed accordingly to reviewer 2 suggestions**:

"P2 L 19. Give some references for these mixing analysis and isotopic studies."

Three references were added in the text and the bibliography. (page 2 line 19)

"P4 L 8. Could the authors comment on the possible effect on the soil/water equilibrium and the associated liberated DOC that is caused by deploying a zero-tension lysimeter?"

A clarification has been added on this point in the section "2.2 water and soil sampling section".
(page 4 line 8-10)

"P5 L23-25. This sentence does not really belong in the methods section."

This sentence has been modified. (page 5 line 26-28)

"P5 L31. This should be Jeanneau et al. (2014). "Jeanneau, L., Jaffrezic, A., Pierson- Wickmann, A.-C., Gruau, G., Lambert, T., and Petitjean, P.: Constraints on the Sources and Production Mechanisms of Dissolved Organic Matter in Soils from Molecular Biomarkers, Vadose Zone J., 13, 2014"

The appropriate reference has been added in the text and in the bibliography. (page 6 line 2)

"P6 L20. Why are the axes called F1 and F2 instead of PC1 and PC2 (principal component). That would make the figures more intuitive to read as well."

In text and figures, F1 and F2 have been replaced by PC1 and PC2.

"P6 L24. The coordinates 'were'."

This sentence has been modified. (page 6 line 31)

[revised manuscript text omitted]